# Study of Insulin Aggregation and Fibril Structure under Different Environmental Conditions

**DOI:** 10.3390/ijms25179406

**Published:** 2024-08-29

**Authors:** Mantas Ziaunys, Kamile Mikalauskaite, Andrius Sakalauskas, Vytautas Smirnovas

**Affiliations:** Institute of Biotechnology, Life Sciences Center, Vilnius University, LT-10257 Vilnius, Lithuania; mantas.ziaunys@gmc.vu.lt (M.Z.); kamile.mikalauskaite@gmc.vu.lt (K.M.); andrius.sakalauskas@gmc.vu.lt (A.S.)

**Keywords:** amyloids, environmental conditions, fibril structure, insulin, protein aggregation

## Abstract

Protein amyloid aggregation is linked with widespread and fatal neurodegenerative disorders as well as several amyloidoses. Insulin, a small polypeptide hormone, is associated with injection-site amyloidosis and is a popular model protein for in vitro studies of amyloid aggregation processes as well as in the search for potential anti-amyloid compounds. Despite hundreds of studies conducted with this specific protein, the procedures used have employed a vast array of different means of achieving fibril formation. These conditions include the use of different solution components, pH values, ionic strengths, and other additives. In turn, this variety of conditions results in the generation of fibrils with different structures, morphologies and stabilities, which severely limits the possibility of cross-study comparisons as well as result interpretations. In this work, we examine the condition–structure relationship of insulin amyloid aggregation under a range of commonly used pH and ionic strength conditions as well as solution components. We demonstrate the correlation between the reaction solution properties and the resulting aggregation kinetic parameters, aggregate secondary structures, morphologies, stabilities and dye-binding modes.

## 1. Introduction

Protein aggregation into amyloid fibrils is associated with the onset and progression of several amyloidoses, including the neurodegenerative Alzheimer’s or Parkinson’s diseases [1]. Despite various anti-amyloid compound screenings and a plethora of studies, there are still only a handful of anti-amyloid compounds available for a limited number of disease-associated proteins [2]. Considering that the incidence of amyloid-related disorders is still rising [3,4], there is a need for a better understanding of the protein aggregation process, which would reduce the number of failed clinical trials conducted on seemingly effective drug molecules [5].

One of the few possible reasons for the lack of effective treatment modalities is the high level of structural variety of the formed amyloid fibrils, which persists both in vivo and in vitro [6,7,8]. It has been observed that amyloid-beta variants [8], Tau protein [9], prion proteins [10] and alpha-synuclein [11], as well as insulin [12], lysozyme [13] and, likely, many other proteins [14] can form fibrils with distinct secondary structures and morphologies. Interestingly, these structural aspects can influence their affinity towards amyloid-specific molecules [12], including aggregation inhibitors [15]. This suggests that structural variability may be an important factor in determining the effectiveness of the tested drug molecules, and it may also explain the high number of failed clinical trials.

In vitro studies of amyloidogenic protein aggregation have demonstrated that environmental conditions have a profound effect on not only the rate of amyloid fibril formation but also their resulting secondary structure and morphology. Since mimicking in vivo conditions would result in unreasonably long protein aggregation times, alterations are often made that greatly accelerate the process [16]. These include changes in protein concentration [17], solution ionic strength [18] and pH values [19], the addition of denaturants, the inclusion of agitation [20] and above-physiological temperatures [21]. In some cases, these differences in conditions are quite extreme, such as prion protein fibrillisation under high concentrations of denaturants [22] or insulin aggregation in acidic solutions [23,24]. Despite being advantageous for quick screening procedures, there is still a lack of information regarding the relation between the various deviations from physiological conditions and the resulting aggregate types.

The aforementioned insulin is a small polypeptide hormone that acts as a regulator of carbohydrate, fat and protein metabolism [25]. Its aggregation into amyloid fibrils is associated with localised amyloidosis at sites of repeat injections [26]. Despite being the cause of a relatively rare disorder, it is often used in protein aggregation studies, especially when analysing amyloid aggregation mechanisms [27,28] and screening for potential anti-amyloid compounds [29,30,31]. While most of these studies are conducted under acidic conditions, a wide variety of different low-pH reaction solutions have been used. These range from simple diluted acids (hydrochloric acid [32,33], acetic acid [34,35]) to buffered solutions (sodium phosphate [36,37], sodium acetate [38,39]). The ionic strength of the reaction solutions also differs significantly between each study, where the addition of sodium chloride is often used to enhance the rate of fibril formation [40,41]. In addition, insulin aggregation studies are often conducted in the presence of various additives, such as nanoparticles [42], lipids [43], organic solvents [44], surfactants [45] and many other compounds [46]. Considering that insulin is known to form multiple distinct fibril types under different conditions [12] and that even relatively small changes in pH [19] or ionic strength [47] can influence their structural variability, this may pose an issue when analysing and comparing data obtained from different studies.

In this work, we examined insulin aggregation under a range of different low-pH conditions and conducted an in-depth analysis of both the fibrillisation process and the resulting fibril structures. The results revealed that there was a highly complex relationship between the reaction solution conditions and the aggregation kinetic parameters as well as with fibril secondary structures and morphologies. These findings highlight the importance of insulin aggregation condition selection and the high level of variability stemming from even small changes in a solution’s ionic strength or pH value.

## 2. Results

The insulin samples prepared under various conditions were first examined using dynamic light scattering (DLS) to determine their dominant oligomeric state. In the case of sodium phosphate solutions with the lowest total ionic strength (Figure 1A), the pH value had minimal effect on the oligomeric state of insulin, and the particle diameter was within the margin of error throughout the entire range (3.2–3.7 nm, comprising monomeric/dimeric insulin [48]). At 300 mM total ionic strength, all the samples had significantly higher particle diameter values compared with using 100 mM NaCl (*p* < 0.01, n = 12), and the solution’s pH value played a more prominent role (Figure 1B). In this case, the pH 2.0-condition insulin particles had the lowest average hydrodynamic diameter (4.0 nm), while this was significantly higher under pH 1.0 (4.4 nm) and pH 3.0 (5.0 nm) conditions. At this ionic strength, the samples were likely composed of dimeric/tetrameric insulin [48]. At 500 mM ionic strength (Figure 1C), the average particle diameter was even higher, with pH 1.0–2.0 having a comparable average value of 4.7–4.9 nm, pH 2.5 having an average of 5.4 nm and pH 3.0 having an average of 5.7 nm. In this case, the largest average-particle-diameter samples likely contained even higher oligomeric states of insulin, such as tetramers or hexamers [48].

Under 20% acetic acid conditions (Ac; Figure 1D) and 25 mM HCl conditions (HCl; Figure 1E), there was also a NaCl-concentration-induced increase in particle diameter. In the case of Ac, the difference was less notable, with the average value ranging from 3.6 nm (100 mM NaCl) to 4.3 nm (500 mM NaCl). Interestingly, despite the pH-meter readout of ~1.8 for Ac conditions, the particle diameter was higher than both pH 1.5 and pH 2.0 sodium phosphate conditions at 100 mM and lower at 500 mM. For HCl, the difference in particle size was considerably more substantial: from 3.5 nm (100 mM NaCl) to 4.8 nm (500 mM NaCl). Unlike Ac conditions, all three ionic strength conditions (with a measured pH value for the HCl solutions of ~1.5) resulted in comparable diameters to the sodium phosphate samples at pH 1.5. This suggests that the presence of 20% acetic acid plays a role in determining the oligomeric state of insulin in solution. A previous study on lysozyme aggregation with high concentrations of organic solvents also displayed its possible role in determining the oligomeric state by affecting the protein’s solvation [44].

Another interesting aspect was that the diameter of insulin had a significant dependence on the solution’s pH value, with the 500 mM ionic strength pH 3.0 sample having the highest particle diameter out of all the conditions. Coincidentally, the solubility of insulin during sample preparation was also noticeably lower under these conditions when compared with other pH values or ionic strengths.

The aggregation of all the different insulin samples was conducted identically as described in the Section 4. During the analysis of kinetic data, it was observed that, despite the majority of aggregation curves possessing a typical sigmoidal shape with a lag phase, exponential signal increase and a plateau (60.5% of a total of 828 curves), there were two different types of curves present (Figure A1). The first type of irregular kinetics had a characteristic double-sigmoidal shape (13.4%), which can be attributed to the formation of ThT-positive insulin aggregation intermediates [49]. The second type of irregular kinetic curves had a gradual and seemingly unending signal increase after the exponential growth phase (26.1%). Such changes in fluorescence intensity could be related to multiple factors, including amyloid fibril maturation [11] (resulting in different ThT binding characteristics), aggregate lateral association and clumping into clusters (ThT entrapment in the structures) or the dissociation of amorphous, ThT-negative structures and their incorporation into amyloid fibrils.

Due to the relatively large number of such irregular curves (39.5%), it was not possible to fit all the data using a standard sigmoidal function. For this reason, the lag time of insulin aggregation was deemed as the time between the start of the reaction and the first notable increase in signal intensity (four times higher signal than the average baseline value). For all three different total ionic strength sodium phosphate solutions (Figure 2A–C), the lag times only experienced minor variations in the majority of samples with pH values from 1.0 to 2.3–2.5, after which they were all considerably higher. Comparing 100 mM (Figure 2A) and 300 mM conditions (Figure 2B), the lag time values overlapped, and the only notable difference was the seemingly higher stochasticity of the 100 mM condition value distribution. Unlike both of these conditions, a total ionic strength of 500 mM resulted in a significantly (n = 12, *p* < 0.05) higher pH 3.0-condition average lag time when compared with lower ionic strength values (Figure 2C). These conditions also had the highest particle hydrodynamic diameters at higher pH values (Figure 1), suggesting a possible correlation between insulin oligomeric states and their lag times in sodium phosphate solutions. In the case of Ac (Figure 2D), higher NaCl concentration conditions resulted in considerably shorter lag time values (300 mM NaCl—70 min; 500 mM NaCl—60 min), when compared with the 100 mM NaCl conditions (130 min). In contrast, under HCl conditions (Figure 2E), the different ionic strengths had the least notable effect, with no significant differences observed between all three NaCl concentrations.

Examination of the apparent rate constants displayed a far more complicated picture, with each set of conditions showing different changes in this parameter. In the case of the 100 mM total ionic strength sodium phosphate solution (Figure 2F), there was a significant increase between the pH 1.0 and pH 1.1 conditions, a gradual decrease between pH 1.1 and pH 1.8, an increase between pH 1.8 and pH 2.6 and a sharp drop between pH 2.6 and pH 3.0. For this ionic strength condition, the apparent rate constant and, to a certain extent, the lag time values at pH 1.0 and from 1.5 to 2.5 could have been influenced by the presence of double-sigmoidal aggregation kinetics (Figure 2K). At a 300 mM ionic strength (Figure 2G), the apparent rate constant gradually increased between the pH 1.0 and pH 2.4 conditions, followed by a sharp decrease between pH 2.4 and pH 3.0. Under these conditions, while the prevalence of double-sigmoidal kinetics was quite low, there was an abundant amount of the second type of irregular curves with an unending signal growth phase (Figure 2L). This phenomenon was highly prevalent at the lowest pH conditions (pH 1.0–1.3) and relatively small at the other marked conditions (Figure 2L), which may explain the low apparent rate constants of these samples. At 500 mM NaCl (Figure 2G), the pH 1.0–1.5 conditions all suffered from a high number of the aforementioned irregular curves (Figure 2M), resulting in a low calculated apparent rate constant. Under higher pH-value conditions (pH 2.0–3.0), the constant had comparable values to the 300 mM NaCl conditions.

Conditions with acetic acid (Figure 2I) had a relatively low number of irregular curves, with the occasional unending plateau at 100 mM and 300 mM NaCl (Figure 2N). The apparent rate constant at 300 mM NaCl was roughly 25% higher than at 100 mM and was within the margin of error for the 500 mM NaCl conditions. The HCl conditions (Figure 2J) resulted in a couple of double-sigmoidal curves at the lowest ionic strength and an abundance of unending plateau kinetics at 300 mM and 500 mM NaCl (Figure 2O). The apparent rate constant under these conditions was lower at higher NaCl concentrations, a factor which may also be linked with the presence of the aforementioned irregular curves. Interestingly, only a few out of all the tested conditions resulted in all twelve aggregation curves having a regular sigmoidal shape. While this does not appear to cause significant variations in the process lag time values, it is very prevalent when determining the apparent rate constants.

In order to determine the condition–structure relationship of insulin amyloid fibrils, the sample FTIR spectra were acquired. To account for possible structural variability among the repeats, all twelve samples from each of the selected conditions were replicated and analysed. The dominant FTIR spectra from each condition were chosen and displayed in Figure 3. The spectra were then compared against each other, and the ones that shared similarities were assigned into variant groups (Figure 3; lowercase letters indicate the group). The five variant FTIR spectra and their second derivative minima positions are displayed in Figure A2.

Under 100 mM total ionic strength conditions, the main maximum position in the insulin FTIR spectra was at 1628 cm^−1^ (related to beta-sheet hydrogen bonding [50]) throughout the entire pH range (Figure 3A). However, between the pH 1.0 and pH 3.0 conditions, the bands at 1659 cm^−1^ and 1640 cm^−1^ became more expressed, which was also visible in the spectra’s second derivatives (Figure 3E). The latter FTIR spectra were nearly identical to previous reports of insulin fibrils generated under pH 2.0 [19]. When the aggregation was conducted under a 300 mM total ionic strength, there was a significantly higher level of variation in the dominant fibril secondary structure (Figure 3B,F). Unlike the 100 mM conditions, the main maximum position ranged from 1628 cm^−1^ to 1632 cm^−1^ at different pH conditions. The pH 1.5 spectrum was identical to the lower-pH-value 100 mM condition spectra, while pH 3.0 spectrum had similarities to the higher-pH-value 100 mM condition spectra, albeit with a higher level of unstructured regions (possibly related to the presence of amorphous aggregates at 1650 cm^−1^ [50]). The remaining 300 mM-condition spectra were different from the 100 mM conditions, while possessing a high level of similarity among each other. This third variant was previously observed when aggregation was conducted under conditions of 100 mM sodium phosphate supplemented with 100 mM NaCl (pH 2.4) [12]. In the case of a 500 mM total ionic strength (Figure 3C,G), the pH 1.0–2.5-condition spectra all shared similarities with minor variations in the main maximum position. The FTIR spectrum of the pH 3.0 sample was comparable to the 300 mM pH 3.0-condition spectrum, with a considerably higher level of noise and more amorphous structures (relatively lower peak in the region associated with beta-sheet hydrogen bonding).

For Ac conditions (Figure 3D,H), the lowest NaCl concentration spectrum shared some similarities to the 100 mM pH 2.5 and 3.0 spectra, with a notable difference in the region associated with turns/loops (second derivative minima at 1663 cm^−1^, as opposed to 1659 cm^−1^). At higher NaCl concentrations, the fibril FTIR spectra had a reduced band at 1641 cm^−1^ and an increase in the 1620 cm^−1^ position. We have previously reported a nearly identical shift in a study that demonstrated a protein-concentration-dependent change in the insulin fibril secondary structure [51]. Here, an increase in NaCl concentration, accompanied by a change in particle hydrodynamic diameter, resulted in the same effect on aggregation as a higher concentration of the protein. Finally, in the case of HCl (Figure 3D,H), the three different NaCl concentration conditions yielded three distinct FTIR spectra, which shared similarities to the spectra of the 100 mM pH 1.0, 100 mM pH 3.0 and 300 mM pH 1.0 fibrils, respectively.

Analysis of insulin aggregate sample atomic force microscopy images (AFM, Figure 4A) revealed that there were three types of fibril distributions in the sodium phosphate solutions of varying ionic strength and pH values. At 100 mM pH 1.5 and pH 2.0, as well as in the 300 mM pH 1.0 conditions, the fibrils formed long intertwined networks, and there was minimal lateral association or aggregate clumping. In contrast, at 300 mM pH 3.0 and 500 mM pH 3.0, very few aggregates could be detected during the AFM scanning procedure. Combined with the FTIR results of these two conditions, it appears that insulin may have formed either amorphous (multiple large clusters were observed on the mica) or unstable aggregates, which would explain the lack of fibrillar structures in the AFM images. Finally, all other conditions resulted in a similar aggregate distribution, with 3–4 µm length fibrils forming lateral associations rather than intertwined networks.

Examining the cross-sectional height (Figure 4B) and width (Figure 4C) distribution of the formed fibrils revealed that the average height varied between 4 nm and 8 nm, while their width ranged from 26 nm to 34 nm. The lowest cross-sectional heights were measured for the 100 mM pH 1.5 and 2.0 and the 300 mM pH 1.0 and pH 3.0 conditions. Three out of these four lowest-height fibrils were also the ones that formed the previously mentioned intertwined networks. The largest average height was determined as belonging to the 500 mM pH 1.0 fibrils, with all other conditions falling in between the two fringe values. The average cross-sectional width was highest for the 100 mM pH 1.0 and 300 mM pH 1.5 conditions and lowest for the 100 mM pH 1.5 and pH 2.0 and 300 mM pH 1.0 conditions, with all other fibrils having intermediate widths.

In the case of insulin fibrils prepared under acetic acid conditions (Figure 5A), the solution ionic strength played a role in determining their morphology. Under 100 mM NaCl conditions, the fibrils were relatively short and laterally associated, while under 300 mM and 500 mM NaCl, they were long and formed intertwined networks. This difference was also evident based on the significantly higher cross-sectional heights (Figure 5B) and widths (Figure 5C) of the Ac-condition aggregates. In contrast, a change in NaCl concentration did not have any significant effect on either the height or width of the insulin fibrils when they were aggregated under HCl conditions, and the morphologies shared similarities to the majority of fibrils under sodium phosphate conditions.

Since the generated fibrils differed in both their secondary structure and morphology, further investigation was dedicated to determining if they also had distinct structural stabilities or dye-binding properties. Analysis of the aggregate stability against denaturation by guanidinium thiocyanate (GuSCN) revealed that most of the aggregates had a similar 1.9–2.1 M denaturation midpoint (Figure 6A–E), with some outliers. The highest stability was observed in the case of the Ac 300 mM and 500 mM NaCl conditions, which corresponded with them having the largest morphological structures. The lowest stabilities were determined for the higher pH and ionic strength conditions, especially for the pH 2.5 and pH 3.0 sodium phosphate solutions. For the 500 mM pH 3.0 condition, it was not possible to calculate the dissociation midpoint due to a very low concentration of stable aggregates.

The dye binding/fluorescence properties (Figure 6F–J) were much more varied, with the highest average quantum yield observed in the case of the Ac 300 mM and 500 mM NaCl conditions (similarly to them having the highest structural stability and fibril size), as well as for the HCl 100 mM NaCl conditions. The lowest average quantum yields of ThT fluorescence were determined for the 100 mM pH 3.0, 300 mM pH 1.5, 500 mM pH 3.0, Ac 100 mM NaCl and HCl 300 mM NaCl conditions, with all other conditions having intermediate values. Considering that there was no clear trend observed under any of the conditions, it is likely that the dye-binding properties of the generated aggregates were structure-specific, as was shown previously [12].

## 3. Discussion

In most insulin aggregation studies, the process of fibril formation is carried out under acidic conditions, with limited regard to the solution’s exact pH or ionic strength values. In this work, we have conducted an examination of insulin amyloid aggregation under a range of different solutions, which encompass most of the commonly used conditions. The results demonstrate that small changes in pH, ionic strength or the solution component type strongly influence the aggregation kinetics, resulting altered fibril secondary structure, morphology, stability and dye-binding properties. These observations exemplify the importance of environmental conditions on in vitro insulin fibrillisation and how the conclusions drawn from the aggregation assays may be influenced by small variations in the chosen reaction solutions.

The first notable aspect is the influence of the reaction solution on the oligomeric state of insulin. Previous reports have demonstrated that insulin may exist in various oligomeric states based on either the solution’s pH or ionic strength conditions [24]. In this study, we observe that both of these factors, as well as the type of the solution’s components, all play a mutual role in determining the association of insulin molecules. While the solution’s ionic strength had a similar influence in increasing the protein’s oligomeric state under all of the chosen pH (1.0–3.0) and solution component (sodium phosphate, acetic acid, hydrochloric acid) conditions, matters were more complex in the case of the other two parameters. Changes in pH values had no effect at low ionic strength conditions, while they had a profound influence under both the 300 mM and 500 mM conditions. The solution components also played a role in determining how the ionic strength can change insulin’s oligomeric state, which is best exemplified in the Ac and HCl solutions. Under hydrochloric acid conditions, the particle hydrodynamic diameter was significantly different among all three NaCl concentrations, while the difference was only minor in the case of acetic acid conditions.

The aggregation kinetics of insulin were also heavily influenced by the type of reaction solution. While the lag times were similar at lower pH conditions, in the case of sodium phosphate solutions, at higher pH conditions, they were significantly longer. Under all three ionic strength conditions, pH 3.0 reaction solutions yielded lag times that were 6 to 10 times larger than their lower-pH-condition counterparts. This may be related to the oligomeric state of insulin and its solubility. During the preparation of the initial samples, it was observed that insulin solubility was particularly low under pH 3.0 conditions. Coupled with the relatively high particle hydrodynamic diameter under these conditions, it may explain why the primary nucleation reactions proceeded at such a slow rate. However, the HCl solutions with 500 mM NaCl displayed similar insulin hydrodynamic diameters without having such a profound effect on lag times, suggesting that the oligomeric state of the protein may not be the most important aspect in this matter.

Another highly peculiar aspect of insulin aggregation was also observed when conducting this large fibrillisation condition screening assay. Under certain sets of pH and ionic strength values, the aggregation reactions resulted in irregular kinetic curves, which were either double-sigmoidal or had gradual unending signal intensity increases after the exponential growth phase. Such irregularities were previously reported for certain specific types of aggregation conditions [49,52]. In this case, we observed that the irregular curves were highly condition-dependent, such as endless plateaus at low pH values and double-sigmoidal curves at intermediate pH values and low ionic strengths. While the double-sigmoidal kinetics can be easily explained by the presence of ThT-positive aggregation intermediates [52], the unending plateau ones can be caused by a number of different factors ranging from fibril maturation to aggregate cluster formation. The relatively high abundance of such irregular curves may influence the aggregation kinetic parameters derived from the data, as there is no definitive way to fit all three curve types using the same kinetic model or function.

Exploring the correlation between aggregation conditions and the secondary structure of insulin aggregates revealed that there were no clearly defined “borders” between aggregate types, with gradual structure–condition changes observed throughout the range of reaction solutions. Analysis of all the dominant secondary structures showed that each one was similar to one of the few previously described structures, with moderate variability in band positions/intensities. The most peculiar FTIR spectra were obtained under pH 3.0 conditions, where the signal to noise ratio was relatively high, and the beta-sheet hydrogen bonding peaks were relatively small, suggesting the presence of a high number of amorphous aggregates. This conclusion was also supported by other assays, including AFM images and the stability assay. It is also worth noting that not a single condition yielded completely identical FTIR spectra and there was a similar level of variability in each case.

Morphological analysis of all the dominant secondary structures indicated that there were, in general, four different classes of morphologies. In most cases, insulin aggregates formed 2–3 µm length structures with a moderate level of lateral association. Fibrils under lower pH and ionic strength conditions assembled into long intertwined structures and had relatively low cross-sectional heights. The largest structures, composed of multiple laterally associated long filaments, formed under higher ionic strength acetic acid conditions. Finally, higher ionic strength and pH conditions yielded low amounts of short fibrils, with most of the aggregates assembling into amorphous structures, as observed in other assays and as indicated by the comparatively low number of structures visible in the AFM images.

The two other aggregate parameters, namely stability against denaturation and dye-binding properties, were closely related to either their structure or morphology. Fibrils that formed larger structures had significantly higher stability against denaturation (best exemplified in the case of Ac 300 mM and 500 mM conditions), while the stability of aggregates formed under higher pH and ionic strength sodium phosphate conditions was either very low or impossible to determine accurately. The average bound-ThT fluorescence quantum yield did not display any clear tendencies with either increasing pH or ionic strength, leading to the conclusion that this parameter was highly dependent on the fibril surface type and ThT binding modes resulting from their distinct secondary structures [12].

In conclusion, this study demonstrates the high level of insulin amyloid aggregation variability and highlights the condition–structure relationships of the resulting fibrils. It shows that even minor alterations throughout the entire tested pH and ionic strength range, as well as the solution component type, can modulate the aggregation kinetic curve type, lag time and apparent rate constant, as well as the resulting aggregate secondary structure, morphology, stability and dye-binding properties. This work encompasses many of the most commonly used insulin aggregation conditions and fills in the gaps between them to create a clearer picture of insulin amyloid formation.

## 4. Materials and Methods

### 4.1. Reaction Solution Preparation

To prepare low-pH-value thermodynamically corrected buffer solutions with identical ionic strengths, the recipes were based on Professor Robert J Beynon’s on-line buffer calculator (http://phbuffers.org/BuffferCalc/Buffer.html (accessed on 25 March 2023)). Phosphoric acid and sodium phosphate (final concentration—100 mM), as well as sodium chloride (final concentrations were determined from the online calculation tool) were dissolved in MilliQ H_2_O at 22 °C. The pH of the solution was determined using a ThermoFisher (Waltham, MA, USA) pH meter (Orion 9110DJWP), and corrections were made using concentrated hydrochloric acid or sodium hydroxide. The solutions were then filtered through 0.22 µm pore-size syringe filters and stored at 4 °C (for up to 24 h before use). The resulting buffer solutions had pH values in the range from 1.0 to 3.0 in 0.1 increments and total calculated ionic strengths of 100 mM, 300 mM and 500 mM.

The other two types of reaction solutions used in this work were 20% acetic acid and 25 mM HCl solutions with different NaCl concentrations. The appropriate volume of 99.5% acetic acid was diluted to 20% using MilliQ H_2_O. Concentrated HCl (25%) was diluted to 25 mM using MilliQ H_2_O. Both solutions were supplemented with either 100 mM, 300 mM or 500 mM NaCl. The solutions were then filtered through 0.22 µm pore-size syringe filters and stored at 4 °C (for up to 24 h before use).

### 4.2. Insulin Aggregation

Human recombinant insulin powder (Sigma-Aldrich, St. Louis, MO, USA, cat. No. 91077C) was dissolved in the previously prepared reaction solutions to a final protein concentration of ~300 µM. The solutions were then filtered using 0.22 µm pore-size syringe filters and the protein concentration was determined by scanning the sample absorbance at 280 nm (ε_280_ = 6335 M^−1^cm^−1^) using a Shimadzu (Kyoto, Japan) UV-1800 spectrophotometer in 3 mm pathlength cuvettes. Thioflavin-T powder (ThT, Sigma-Aldrich, cat. No. T3516) was dissolved in Milli-Q H_2_O to a concentration of ~11 mM, filtered through a 0.22 µm pore-size syringe filter, diluted to 10 mM (ε_412_ = 23,250 M^−1^cm^−1^) and stored at −20 °C under dark conditions. The insulin solutions were then combined with ThT and their respective reaction solutions to result in mixtures with 100 µM ThT and 200 µM insulin. To prevent batch-to-batch variability, all protein solutions were prepared simultaneously using the same insulin powder batch. The solutions were then frozen at −80 °C and thawed prior to the aggregation experiments.

The insulin reaction solutions were distributed into 96-well plates (Greiner, Kremsmunster, Austria, cat. No. 11877192; final volume in each well—200 µL) in an alternating placement and avoiding corner wells to account for the plate-edge-effect during the aggregation process [53] (an example is shown in Figure A3). The plates were then sealed using Nunc sealing tape (ThermoFisher, Waltham, MA, USA, cat. No. 10265411) and incubated in a ClarioStar Plus plate reader (BMG Labtech, Ortenberg, Germany) with a constant 60 °C temperature and no agitation. Sample measurements were taken every 5 min using 440 nm excitation and 480 nm emission wavelengths (0.1 s settling time, 50 flashes per sample). For each condition, a total of 12 samples were aggregated (4 samples per run). After aggregation, the plates were cooled down to room temperature prior to further use. Due to the sample incubation being at an elevated temperature, small deviations from the initial reaction conditions (pH, ionic strength and protein concentration) may have occurred during the procedure.

### 4.3. Dynamic Light Scattering

The non-aggregated insulin samples (200 µM protein concentration, 20 µL volume) were placed in 1 mm pathlength quartz cuvettes and covered with plug caps. The protein hydrodynamic diameter was then measured using a Malvern Panalytical (Malvern, UK) Zetasizer µV light-scattering detector at a constant 22 °C temperature. For each sample, ten technical replicates (each composed of ten scans over the span of 1 min) were measured. The thermal equilibration time was set to 1 min, and the total scan time was 10 min.

### 4.4. Kinetic Data Analysis

Under multiple conditions, the aggregation kinetic data could not be fit using standard sigmoidal curves, due to either requiring double-sigmoidal kinetic curves or there being a gradual shift in the signal intensity after the main exponential growth phase. For this reason, the extent of the lag phase was determined as the time period, during which the sample signal intensity remained below a set intensity value for each sample (average value of the initial baseline (10 points) multiplied by 4 (to exclude random signal variations)). An example of the lag time determination is show in Figure A4. To obtain the maximum apparent rate constants, first-order derivatives were calculated for each reaction curve as described previously [52]. The maximum point value from each derivative was then divided by the signal intensity at the end of the reaction (the end-point fluorescence intensity was determined at the same time for each set of conditions; an example is shown in Figure A4).

### 4.5. Fourier-Transform Infrared Spectroscopy (FTIR)

Aliquots of 100 µL were taken from each sample and centrifuged at 10,000 RPM (Fisherbrand™ High-Speed Mini-Centrifuge, ThermoFisher, Waltham, MA, USA, cat. no 12972041) for 15 min. The supernatants were then removed, and the aggregate pellets were resuspended into 200 µL D_2_O with 400 mM NaCl (addition of NaCl improves insulin fibril sedimentation [54]). The centrifugation and resuspension procedures were repeated an additional two times. After the final centrifugation, the aggregate pellets were resuspended into 40 µL D_2_O with 400 mM NaCl, resulting in a final protein concentration of ~500 µM (assuming the majority of insulin was in its aggregated state). The sample FTIR spectra were acquired and analysed as described previously [55]. In short, the fibril FTIR spectra were scanned using an Invenio S infrared spectrometer (Bruker, Billerica, MA, USA). For every sample, 256 interferograms were recorded at a 2 cm^−1^ resolution and averaged. The D_2_O and water vapour spectra were then subtracted from the sample spectra, which were then baseline-corrected and normalised to the same Amide I/I’ band area (1700–1595 cm^−1^). Data analysis was performed using the GRAMS software Version 9.3 SP1, (ThermoFisher, Waltham, MA, USA).

### 4.6. ThT-Binding Assay

Samples from each set of conditions were chosen based on which FTIR spectra were the most common among the 12 repeats. The samples (100 µL) were then combined with their initial reaction solutions (900 µL) and incubated at 60 °C for 24 h in non-binding 2.0 mL volume test-tubes to increase the number of fibrils for further experimental procedures. To mitigate the effect of solution pH and ionic strength on the dye-binding parameters, the samples were centrifuged as described previously, and the aggregates were resuspended into an equivalent volume of 100 mM sodium phosphate (pH 2.0, adjusted to a total ionic strength of 300 mM using NaCl).

Aliquots of 200 µL of each sample were supplemented with 2 µL of a 10 mM ThT stock solution, mixed and incubated at 22 °C for 1 h. The samples were then centrifuged at 10,000 RPM for 10 min and resuspended into 200 µL of the pH 2.0 buffer solution, which did not contain ThT. This centrifugation and resuspension procedure was repeated 3 times to remove non-bound ThT molecules from the solution. After the final resuspension step, the aggregate samples were placed in 3 mm cuvettes, and their absorbance spectra were scanned using a Shimadzu UV-1800 spectrophotometer (200–600 nm range, 1 nm steps). The fluorescence of bound-ThT molecules was scanned using a Varian CaryEclipse spectrofluorometer (Agilent Technologies, Santa Clara, CA, USA, 440 nm excitation, 480 nm emission wavelengths, 2.5 s signal averaging time). For each sample, three technical replicates were scanned, and the spectra were averaged.

The absorbance spectra were baseline-corrected between 300 nm and 600 nm using the Origin Software 2018 (OriginLab, Northampton, MA, USA). Baseline correction function (B-spline with 4 anchor points at each end of the spectrum). The spectra were then integrated between 340 nm and 520 nm to obtain the areas of the bound-ThT absorbance spectra. ThT fluorescence intensities were corrected for the inner filter effect using their respective absorbance values at 440 nm and 480 nm as described previously [56].

### 4.7. Atomic Force Microscopy (AFM)

Aliquots of each sample were diluted to a 40 µM protein concentration, placed on freshly cleaved mica and incubated at room temperature for 5 min. The micas were then gently washed with 2 mL of H_2_O and dried using airflow. The AFM image acquisition procedure was performed as described previously [55] using a Dimension Icon atomic force microscope (Bruker, Billerica, MA, USA). The AFM images were analysed using the Gwyddion 2.57 software (http://gwyddion.net, accessed on 28 December 2020). From each image, 30 fibril cross-sectional heights and widths were determined by tracing lines perpendicular to the fibril axes. The lines were only traced on fibril ends, which were not part of aggregate clusters, to reduce the effect of lateral association on the calculated height and width parameters. Data were compared using one-way ANOVA Bonferroni means comparison (*p* < 0.01, n = 30).

### 4.8. Aggregate Denaturation Assay

Aliquots of each sample (46 × 10 µL) were combined with a range of guanidinium thiocyanate (GuSCN) solutions of different concentrations (90 µL of pH 2.0, 100 mM sodium phosphate; total ionic strength without GuSCN was adjusted to 300 mM using NaCl). The final sample GuSCN concentrations ranged from 0 to 4.5 M in 0.1 M increments. The prepared solutions were then incubated at 22 °C for 24 h, after which they were placed into 96-well non-binding plates (cat. No 3881, Fisher Scientific, Hampton, NH, USA, final volume was 100 µL in each well). The sample optical densities at 600 nm were scanned using a ClarioStar Plus plate reader (BMG Labtech, Ortenberg, Germany) with 0.1 s settling time and 50 flashes per sample. Each sample was scanned three times and the plates agitated for short periods in between (10 s, 600 RPM orbital agitation). The data was fitted using a Boltzmann sigmoidal equation to determine the mid-point of insulin aggregate denaturation (Figure A5).

## Figures and Tables

**Figure 1 ijms-25-09406-f001:**
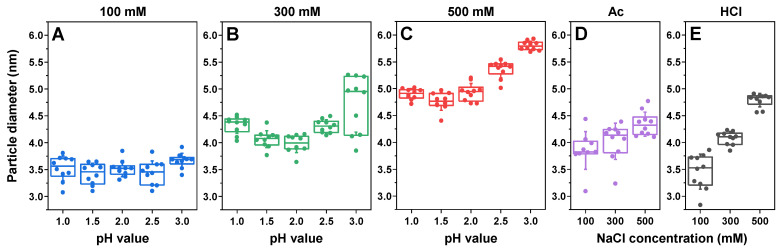
Insulin sample dynamic light scattering (DLS) measurements. The dependence of the insulin hydrodynamic diameter on the sample pH value in sodium phosphate solutions of 100 mM (**A**), 300 mM (**B**) and 500 mM (**C**) total ionic strength. Dependence of the insulin hydrodynamic diameter on the sample ionic strength in 20% acetic acid (**D**) and 25 mM HCl (**E**) solutions. Box plots indicate the interquartile range, and the error bars represent one standard deviation (n = 10).

**Figure 2 ijms-25-09406-f002:**
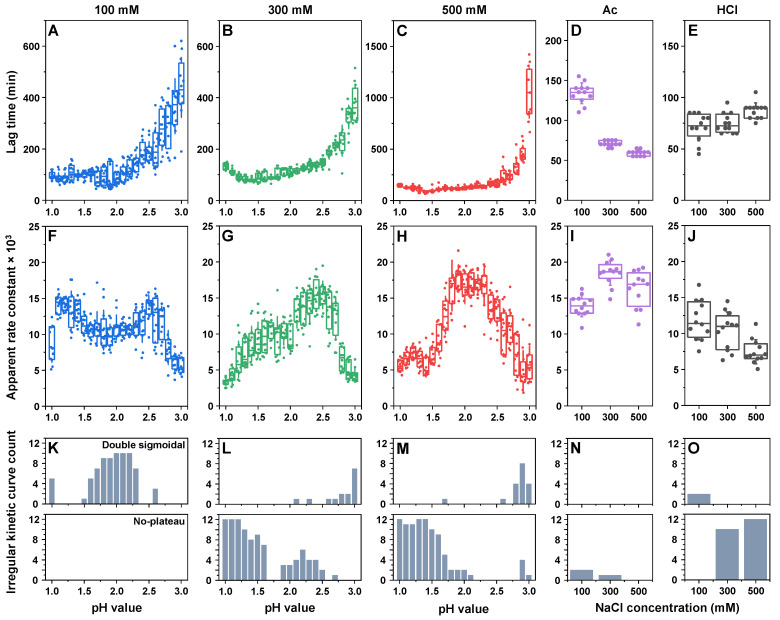
Insulin aggregation kinetic parameters and curve types. The lag time and apparent rate of insulin aggregation under different pH conditions in sodium phosphate solutions of 100 mM (**A**,**F**), 300 mM (**B**,**G**) and 500 mM (**C**,**H**) total ionic strength. The lag time and apparent rate of insulin aggregation under different NaCl concentrations in 20% acetic acid (**D**,**I**) and 25 mM HCl (**E**,**J**) solutions. The distribution of double-sigmoidal (upper panel) and unending signal increase (lower panel) curves for 100 mM (**K**), 300 mM (**L**), 500 mM (**M**) total ionic strength sodium phosphate solutions, as well as for 20% acetic acid (**N**) and 25 mM HCl (**O**) solutions. Box plots indicate the interquartile range, and the error bars represent one standard deviation (n = 12).

**Figure 3 ijms-25-09406-f003:**
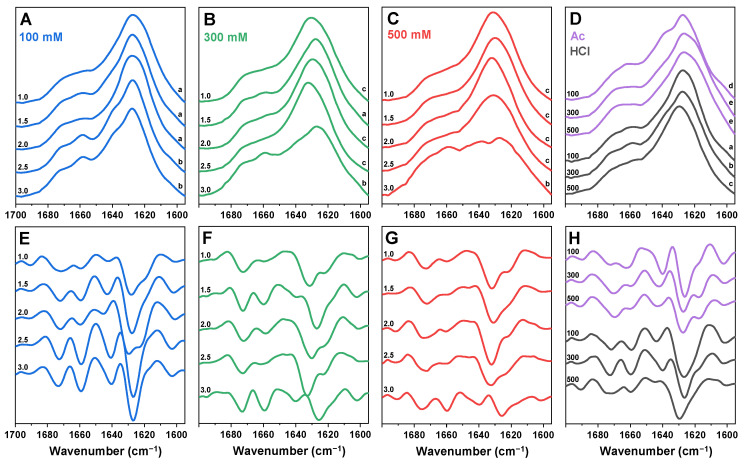
Comparison of insulin aggregate secondary structures. FTIR spectra of insulin aggregates formed in sodium phosphate solutions of 100 mM (**A**), 300 mM (**B**) and 500 mM (**C**) total ionic strength (pH values are indicated on the left sides of the spectra). FTIR spectra of aggregates formed in Ac and HCl conditions (**D**; ionic strengths are indicated on the left side of the spectra). Second derivatives of the FTIR spectra (**E**–**H**) are colour-coded and displayed below their respective FTIR spectra. FTIR spectra variants are indicated by lowercase letters on the right side of the spectra.

**Figure 4 ijms-25-09406-f004:**
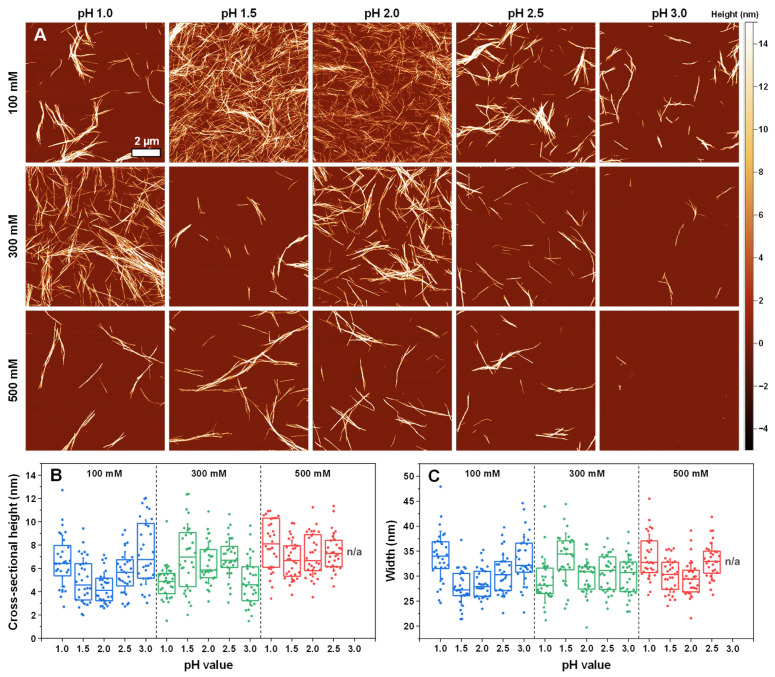
Morphology of insulin aggregates prepared in sodium phosphate solutions. The atomic force microscopy (AFM) images of different condition insulin aggregates (**A**) and their cross-sectional heights (**B**) and widths (**C**). For each condition, 30 traces were taken to determine the height and width values. Box plots indicate the interquartile range, and the error bars represent one standard deviation.

**Figure 5 ijms-25-09406-f005:**
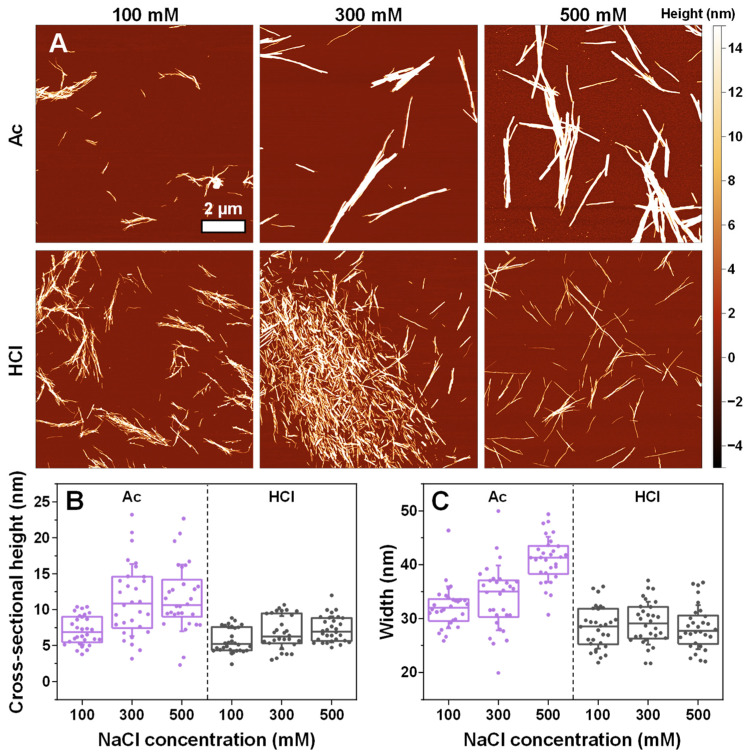
Morphology of insulin aggregates prepared in 20% acetic acid (Ac) and 25 mM hydrochloric acid (HCl) solutions. The atomic force microscopy (AFM) images of insulin aggregates under different conditions (**A**) and their cross-sectional heights (**B**) and widths (**C**). For each condition, 30 traces were taken to determine the height and width values. Box plots indicate the interquartile range, and the error bars represent one standard deviation (n = 30).

**Figure 6 ijms-25-09406-f006:**
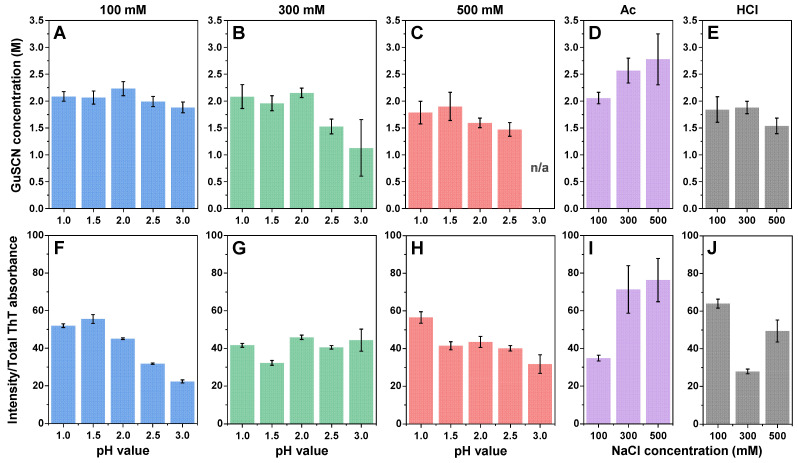
Insulin aggregate stability and bound-ThT fluorescence quantum yield. The midpoint of the aggregate denaturation curves of insulin aggregate samples prepared under 100 mM (**A**), 300 mM (**B**), 500 mM (**C**) total ionic strength sodium phosphate conditions, as well as under acetic acid (**D**) and hydrochloric acid (**E**) conditions. The average bound-ThT fluorescence quantum yield of insulin aggregate samples prepared under 100 mM (**F**), 300 mM (**G**), 500 mM (**H**) total ionic strength sodium phosphate conditions, as well as under acetic acid (**I**) and hydrochloric acid (**J**) conditions. For each condition, three measurements were taken and averaged. The error bars represent one standard deviation (n = 3).

## Data Availability

All data are available in the Appendix A.

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
