# Peer review of "Study of Insulin Aggregation and Fibril Structure under Different Environmental Conditions"

_ijms, 2024, doi:10.3390/ijms25179406_

Round 1
Reviewer 1 Report
Comments and Suggestions for Authors
The manuscript addresses a very relevant topic, which is the study of the influence of the environment on the formation of amyloid fibrils. The manuscript contains scientifically interesting results. I recommend approval subject to revisions.
-The introduction of the manuscript is well-written. However, more recent studies could be added to contextualize the work done in the area of amyloid aggregates in different environments. The most recent manuscript cited is from 2022. There are several other recent studies addressing the topic (example: 10.3390/molecules28196891).
-In lines 99 and 100, the authors wrote, “…This suggests that the presence of 20% acetic acid plays a role in determining the oligomeric state of insulin in solution…”. Did you attempt to use computational methods to understand the interactions that favor this? The article 10.3390/molecules28196891 covers the subject well. Perhaps the work could move in this direction.
-In the experiments using ThT, no control was performed showing that the probe with insulin and other isolated solution components does not fluoresce, only when fibrils are formed.
-The AFM images (Figures 4 and 5) show the overall content of the aggregates but do not show the morphology in detail. Would it not be possible to conduct transmission electron microscopy experiments?
-I did not find information in the methodology section indicating that the DLS, infrared, and fluorescence data were repeated. Did the authors repeat these experiments? This information should be added.
-A section concluding the data should be added to the manuscript.
Comments on the Quality of English Language
Minor editing of English language required.
Author Response
Reviewer #1
The manuscript addresses a very relevant topic, which is the study of the influence of the environment on the formation of amyloid fibrils. The manuscript contains scientifically interesting results. I recommend approval subject to revisions.
-The introduction of the manuscript is well-written. However, more recent studies could be added to contextualize the work done in the area of amyloid aggregates in different environments. The most recent manuscript cited is from 2022. There are several other recent studies addressing the topic (example: 10.3390/molecules28196891).
Thank you for the suggestion. We have added references to additional recent studies and expanded on their content in the Introduction and Results.
-In lines 99 and 100, the authors wrote, “…This suggests that the presence of 20% acetic acid plays a role in determining the oligomeric state of insulin in solution…”. Did you attempt to use computational methods to understand the interactions that favor this? The article 10.3390/molecules28196891 covers the subject well. Perhaps the work could move in this direction.
We did not use computational methods in this study, however we have included a reference to the study in the manuscript’s Result section. The high concentration of organic solvent may determine the oligomeric state of insulin by affecting the protein’s solvation. The 20% acetic acid solution did not appear to have a similar effect on insulin lag time as acetone did in the lysozyme aggregation study.
-In the experiments using ThT, no control was performed showing that the probe with insulin and other isolated solution components does not fluoresce, only when fibrils are formed.
We have conducted additional measurements with the different reaction solutions and ThT (before and after 48 hour incubation at 60°C) to determine if any of the components cause a significant increase in ThT fluorescence intensity. The changes in ThT intensity were minimal and several orders of magnitude lower than the increase caused by insulin fibrils.
In the case of solutions with insulin, its aggregated form produces several orders of magnitude higher ThT fluorescence intensity than the non-aggregated state. This is shown in the raw kinetic data, provided as supplementary material.
-The AFM images (Figures 4 and 5) show the overall content of the aggregates but do not show the morphology in detail. Would it not be possible to conduct transmission electron microscopy experiments?
Thank you for the suggestion. Regular TEM images would provide similar or, in some cases, less information regarding the morphological details of the fibrils, such as cross-sectional height. We intend to use this study to select conditions for future research of the fibril structure with Cryo-TEM, however, solving the structure of such a vast array of distinct fibrils will require a significant amount of time.
-I did not find information in the methodology section indicating that the DLS, infrared, and fluorescence data were repeated. Did the authors repeat these experiments? This information should be added.
We have added additional information regarding the repeats and noted that they were technical replicates of the same sample batch.
-A section concluding the data should be added to the manuscript.
The manuscript contains a conclusion paragraph as the last part of the discussion. We believe this is a more suitable placement, as it follows a detailed discussion of the results. Based on the journal format, a conclusion section is optional and would follow the Materials and Methods section, which may not be optimal placement for this manuscript.
Reviewer 2 Report
Comments and Suggestions for Authors
In this manuscript Ziaunys and coauthors explore the influence of moderate-high salt concentration and pH changes (in extremely low range) on the in vitro kinetic of insulin fibrils formation.
The work is clearly explained and the authors document their conclusions with plentiful of experimental results. However, some minor concerns need to be addressed before the manuscript is ready for publication.
In the kinetic experiments, starting insulin concentration is fixed to 200 µM in all experimental conditions, varying pH and/or ionic strength. Incubation at 60 °C.
1. Incubation at 60 °C may provoke solvent evaporation, leading to an higher ionic strength and protein concentration increasing compared to the starting experimental conditions. Have you checked for proper volume in each well at the end of the incubation?
2. Have you checked for each condition tested if the pH value is still the same at the experiment endpoint?
Author Response
Reviewer #2
In this manuscript Ziaunys and coauthors explore the influence of moderate-high salt concentration and pH changes (in extremely low range) on the in vitro kinetic of insulin fibrils formation.
The work is clearly explained and the authors document their conclusions with plentiful of experimental results. However, some minor concerns need to be addressed before the manuscript is ready for publication.
In the kinetic experiments, starting insulin concentration is fixed to 200 µM in all experimental conditions, varying pH and/or ionic strength. Incubation at 60 °C.
- Incubation at 60 °C may provoke solvent evaporation, leading to an higher ionic strength and protein concentration increasing compared to the starting experimental conditions. Have you checked for proper volume in each well at the end of the incubation?
We agree that solution evaporation at such temperatures might be an issue. For this reason, we have taken every possible precaution to minimize the effect. The 96-well plate was sealed with Nunc sealing tape, which is designed to prevent evaporation from the wells. We also avoided using the plate corners, as they are known to sometimes seal improperly (Appendix Figure A3). The incubation time was also selected to be as low as possible and the reactions were concluded when all kinetic curves reached a plateau.
We did not notice any significant level of evaporation in any of the wells during this study.
- Have you checked for each condition tested if the pH value is still the same at the experiment endpoint?
We have conducted an additional investigation into this matter and observed that the low level of sample evaporation can lead to small changes in sample pH values (up to 0.1 deviations).
We have added an additional statement in the Materials and Methods Insulin aggregation section that the solution conditions may become slightly altered due to the incubation procedure.